# TokenAR: Multiple Subject Generation via Autoregressive Token-level enhancement

## Abstract

Autoregressive Model (AR) has shown remarkable success in conditional image generation. However, these approaches for multiple reference generation struggle with decoupling different reference identities. In this work, we propose the *TokenAR* framework, specifically focused on a simple but effective token-level enhancement mechanism to address reference identity confusion problem. Such token-level enhancement consists of three parts, 1). Token Index Embedding clusters the tokens index for better representing the same reference images; 2). Instruct Token Injection plays as a role of extra visual feature container to inject detailed and complementary priors for reference tokens; 3). The identity-token disentanglement strategy (ITD) explicitly guides the token representations toward independently representing the features of each identity. This token-enhancement framework significantly augments the capabilities of existing AR based methods in conditional image generation, enabling good identity consistency while preserving high quality background reconstruction. Driven by the goal of high-quality and high-diversity in multi-subject generation, we introduce the *InstructAR Dataset*, the first open-source, large-scale, multi-reference input, open domain image generation dataset that includes 28K training pairs, each example has two reference subjects, a relative prompt and a background with mask annotation, curated for multiple reference image generation training and evaluating. Comprehensive experiments validate that our approach surpasses current state-of-the-art models in multiple reference image generation task. The implementation code and datasets will be made publicly.

## 1 Introduction

Large-scale autoregressive models (AR)(Shao et al., 2025; Yu et al., 2021; 2022; Han et al., 2025; Tian et al., 2024) have demonstrated impressive performance in images generation. Recently, AR-based methods(Mu et al., 2025; Chen et al., 2025; Sun et al., 2024; Wu et al., 2025) have expanded the generation capabilities from random generation to conditional generation, encompassing text-to-image generation, style transformation, and image editing, utilizing complex attention mechanisms or intricate loss functions. However, these methods (Mu et al., 2025; Chen et al., 2025) are mostly designed for single-image editing, and their performance on multiple reference condition-based tasks is less than satisfactory. Current AR methods often fail to distinguish multiple identities when more than one reference subject is provided. This deficiency in multiple reference generation seriously impedes the generalization of the model's capabilities to tasks with more conditions.

Challenges for multiple reference generation lie in both the dataset and the model. We identify several issues in current open-source datasets for the conditional image generation task: *1) Insufficient dataset size*: Current conditional image datasets typically have less than 10k data records, and only a few of them have a proper description about the target image. Additionally, there is a lack of clear generation scripts in those studies that propose automatic data generation strategies. *2) Lack of pose difference:* Datasets gathered from real-world images often suffer from difficulty in taking different poses of the same subject. Besides, segmented data pairs lead to insufficient subject pose transformation between reference images and target images. *3) Deficiency in region mask:* There is a significant problem annotating an appropriate mask to target images in the existing dataset, for most images have complex backgrounds, which seriously affects the performance of segmentation models. Low-logits regions also make the subject extraction process more precise. *4) Absence*

*of relation diversity:* Existing datasets often focus on a single static subject, with few interactions between subjects in the data.

To address the above deficiencies, we propose a new dataset, termed ***InstructAR Dataset***, for multiple reference generation. The construction pipeline shown in Figure 1 involves three stages: ***1) Comprehensive relation generation:*** Two reference images and a relation are sampled as conditions on the relation-based DiT(Qingyu Shi, 2025) to yield target images with two subjects in the relation. There are 11 kinds of relation, 44 subject categories, and more than 18k human pictures(Sec. 2.2). ***2) Fine-grained part mask annotation:*** This stage involves a combination of automatic and manual processes. We automatically generate masks for foreground in target images through an open-source model(Meyer & Spruyt, 2025) and manually set the logits threshold to maintain reasonable background regions (Sec. 2.1). ***3) Context-based and visual-based filter:*** We use VLM(Team, 2025) with in-context learning method to drop those data whose target images mismatch relative descriptions, and filter those whose foreground has little similarity with reference subjects via DINOv2 scores(Oquab et al., 2023) (Sec. 2.1). This procedure yields data for a multiple reference generation dataset (with two reference subjects). Figure A3 shows more examples.

With the constructed dataset, several key questions must be thoroughly examined for the framework design: ***1)*** Current AR models struggle with multiple identities preservation in the generation process due to low-resolution global feature loses detailed token representations of each identity. ***2)*** For multiple subjects as input, models cannot decouple tokens from different identities since only position embedding can help transformer layers distinguish tokens from different subjects, which makes the training process difficult to learn the connection between tokens belonging to one subject. ***3)*** Although AR models share a similar model structure with LLMs, the importance of tokens is ignored in recent research.

To this end, we proposed the **TokenAR** framework. For multiple identities confusion, we introduce ***Token Index Embedding*** as index-level position instruction to assist the attention mechanism in being more focused on tokens belonging to one subject (Sec. 3.1.2). To preserve the consistency of multiple identities, we adapt the training and inference strategy to ***Identity-token Disentanglement Strategy***. This approach fully utilizes the input token information and the loss function forces the token representations to maintain high-frequency details of each subject during feature passing to deeper layers (Sec. 3.1.2). Furthermore, inspired by the impressive performance of soft prompts in LLMs,we introduce ***Instruct Token Injection***, which serves as an extra visual feature container to inject detailed and complementary priors for the reference tokens, thereby enhancing the overall generation quality (Sec. 3.1.3). In summary, our contributions are threefold:

- We introduce a novel multiple reference dataset–**InstructAR Dataset**, the first open-source, multi-subject, open-domain dataset, specifically designed for multiple reference generation.

- We propose **TokenAR**, a framework for multiple subject generation combined with an effective token-level enhancement mechanism, enhancing the AR model's capabilities in subject decouple and identity preservation.

- Extensive experiments demonstrate that our proposed method achieves superior identity preservation and surpasses current state-of-the-art models for multi-ID conditional generation tasks.

## 2 INSTRUCTAR DATASET

### 2.1 DATASET CONSTRUCTION

To facilitate robust training and evaluation for multi-reference generation, we introduce the InstructAR Dataset. Its construction pipeline is specifically designed to overcome four critical deficiencies in existing datasets: limited scale, insufficient pose variation, inaccurate masks, and a lack of quality control. To address scale and diversity, we employ a relation-guided generative model (Qingyu Shi, 2025) to synthesize a vast and varied image corpus. For precise annotations, we use BNE2 (Meyer & Spruyt, 2025) for automated mask generation, followed by manual refinement. Finally, to ensure quality, we institute a rigorous two-stage filtering protocol, using a Vision-Language Model (Team,

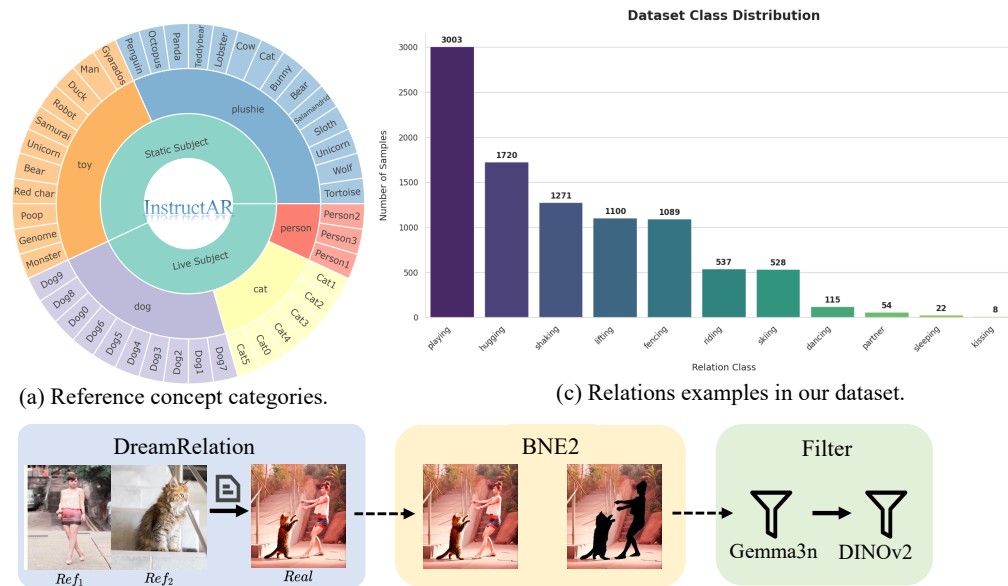

(a) Reference concept categories.

(c) Relations examples in our dataset.

(b) Dataset construction process.

Figure 1: **Overview of InstructAR dataset**, (a) Visualization of reference concept categories. Covering diverse reference concepts to insert including: toys, dogs, cats, persons, and plushies, (b) Visualization of relations in part of our datasets. There are 11 kinds of relations in total, with playing, hugging, and shaking the most, (c) Dataset construction process which contains three stages.

Table 1: **Comparison of different datasets.** Compared with three other different open source datasets, our new dataset has more samples, with multiple reference images, and the pose of reference subjects is completely different from that in target images.

| Datasets | Multiple Reference | Automatic Generated | Open Domain | #Edits | #Reference Number | Source Example | Prompt | Target Example |
|---|---|---|---|---|---|---|---|---|
| DreamBooth | ✗ | ✗ | ✓ | 342 | 1 | | *A backpackdog backpack near the window.* | |
| SpatialSubject200K | ✗ | ✓ | ✓ | 20399 | 1 | | *It is positioned upright in a grassy field with sunlight illuminating its surface.* | |
| SubjectDataset10K | ✓ | ✗ | ✗ | 9988 | 2-3 | | *A close up of a red car parked on a road near a field.* | |
| InstructAR(Ours) | ✓ | ✓ | ✓ | **28,027** | 2 | | *Plushie tortoise and person are cooking together.* | |

2025) to validate semantic alignment and DINOv2 (Oquab et al., 2023) feature similarity to confirm identity preservation. Shown in Figure 1, the detailed construction process is described below:

1) Source Material Curation. Our process begins by synthesizing the training data. We first curate a pool of reference subjects from the DreamRelation Benchmark (Qingyu Shi, 2025) and a human parsing dataset (Liang et al., 2015). For each sample, we draw two reference images ($Ref_1, Ref_2$) and a textual relation. These components are then input to the DreamRelation model (Qingyu Shi, 2025) to generate a corresponding target image, which we designate as our ground-truth, $Real$. This generative approach allows us to create a large-scale dataset with diverse and specific multi-subject interactions.

2) Foreground and Background Extraction: We use the BNE2(Meyer & Spruyt, 2025) tool to separate the foreground and background from the ground truth image, $Real$, resulting in two new images: $foreground$ and $background$. The $foreground$ image represents the primary foreground object, while the $background$ image contains only the background context.

3) Semantic and Identity Validation via VLM-based Filtering. A significant challenge in data curation is ensuring semantic fidelity and subject integrity. To this end, we introduce a rigorous filtering stage using a powerful Vision-Language Model, Gemma-3n-E4B-it (Team, 2025). Through a structured prompt, we task the model with performing two validation checks on each sample. Any sample

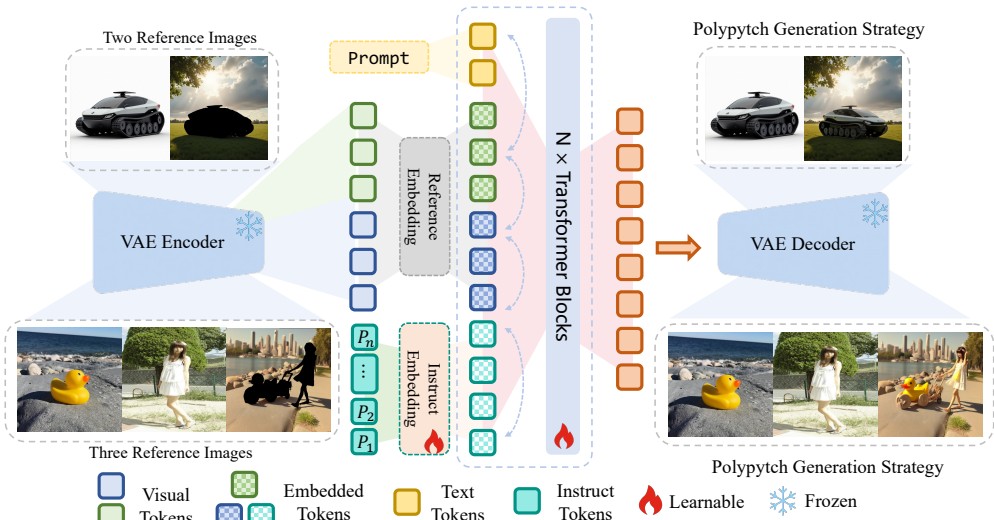

Figure 2: **Overview of TokenAR.** Given different number of reference image, our approach process concatenation of multiple reference and a masked background through a frozen VAE Encoder to get visual tokens keeping their details. Text input, prompt, is encoded via text encoders to extract text guidance information. All the tokens are combined with instruct tokens $P_i$ together and fed into the learnable transformers blocks for identity-token disentanglement image generation, enabling good identity consistence and background reconstruction.

that fails to pass both of these programmatic checks is subsequently discarded. This automated validation process is crucial for maintaining a high-quality, instruction-aligned training corpus.

4) DINOv2 Similarity Filtering: Finally, we perform a similarity check using the DINOv2(Oquab et al., 2023). We filter the collected images to ensure a high degree of visual similarity between the foreground image, $foreground$, and both reference images, $Ref_1$ and $Ref_2$. Specifically, we keep only the samples where the DINOv2 feature similarity between $foreground$ and $Ref_1$ is greater than a predefined threshold $\delta$, and the DINOv2 feature similarity between $foreground$ and $Ref_2$ iss also greater than $\delta$.

$$Valid = \min(Sim_{dino}(foreground, Ref_1), Sim_{dino}(foreground, Ref_2)) \geq \delta \qquad (1)$$

This strict filtering process guaranteed that the reference images were highly relevant to the foreground object we aimed to generate.

## 2.2 DATASET OVERVIEW

Table 1 presents a detailed comparison between our proposed InstructAR dataset and existing benchmarks. InstructAR is divided into a training set of 28,027 samples and a test set of 203 samples. Each sample is a comprehensive tuple, consisting of two reference subject images, a segmented background, and a corresponding textual description of the target scene. To ensure diversity, the reference inputs were curated by sampling objects from DreamRelationBench (Qingyu Shi, 2025) (303 images across 44 categories) and portraits from a large-scale human parsing dataset (Liang et al., 2015) (17,706 distinct images).

## 3 METHODOLOGY: TOKENAR

### 3.1 RESEARCH DEFINITION

This work extends multiple reference generation research in AR models to token level research. Given a Text prompt $\mathcal{T}$ and a group of reference images $\mathcal{I} = \{I_i\}_n$, the AR model generates token sequence $\mathbf{q} = \{q_i\}_m$ including target images information. We use the image editing framework based on EditAR(Mu et al., 2025) as a foundation to develop TokenAR model for fine-grained identities preservation, as illustrated in Figure 2.

### 3.1.1 MOTIVATION OF DIFFERENT COMPONENTS

**Index-level *vs*. Position-level.** Positional embeddings like RoPE are insufficient for multi-reference generation. While RoPE handles token positions, it is source-agnostic and provides no information to group tokens originating from different reference images. This is a critical flaw for identity preservation, as the model cannot easily distinguish which tokens belong to which subject. To solve this, we introduce a learnable Token Index Embedding. This embedding explicitly clusters tokens by their source, assigning a unique, shared vector to all tokens from the same reference image. This provides the necessary group-level signal for the model to differentiate and preserve multiple distinct identities.

**Only target *vs*. Keeping all.** Compared to the generating all available information (*e.g.,* reference images), origin aim of generation task (only generate target images) demands lower precision in information preservation, requiring little training steps to achieving it. However, when it comes to the preservation of multiple identities, detailed information of each subject is required for the characteristics of different subjects merged together, which impedes the accuracy of generation. Therefore, we introduce the Identity-token Disentanglement Strategy, which leverages the reference tokens as a dense supervisory signal, compelling the model to reconstruct target tokens accurately. This reconstruction objective explicitly guides the token representations toward independently representing the fine-grained features of each identity.

**Extra tokens to guide specific task.** In recent LLMs studies, in-context learning plays important role in multiple task generalization. Extra constructed tokens acted as guidance assists the performance of LLMs to get remarkable performance in different tasks. However, although AR models share similar model structure with LLMs, few researches explore potential of such instructive tokens. Thus, we incorporate trainable Instruct Tokens and origin tokens to introduce instructive guidance for this task and containing additional information of other guidance tokens during generation.

### 3.1.2 TOKEN INDEX EMBEDDINGS AND IDENTITY-TOKEN DISENTANGLEMENT STRATEGY

Conventional position embeddings like Rotary Position Embedding (RoPE) preserve relative positional information but are agnostic to the source of tokens. They lack an intrinsic mechanism to distinguish or group tokens originating from different reference subjects. To address this, we introduce a learnable Token Index Embedding. This assigns a unique, learnable vector to all tokens from the same source image, acting as an explicit grouping signal. This allows the model to differentiate token sources, a crucial prerequisite for preventing identity confusion in multi-subject generation.

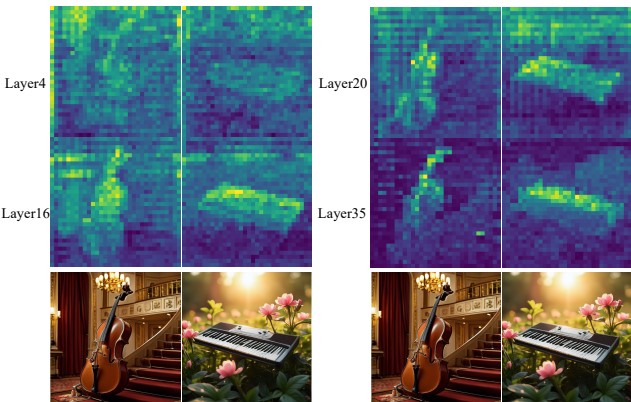

Figure 3: **Average cross-attention map** of target image in different transformer layers.

Furthermore, preserving high-frequency, identity-specific details remains a central challenge. Mainstream approaches often resort to complex architectural modifications, such as specialized encoders (Mu et al., 2025; Wu et al., 2025), or alter the core attention mechanism (Chen et al., 2025). These methods can be inefficient or create discrepancies with the base model's pre-training, leading to training instability. In contrast, our Identity-token Disentanglement Strategy, complemented by the Token Index Embedding, offers a more direct solution. During training stage, we employ concatenation consisting of processed reference images $\{I_i\}_m : I_{concat} = [I_1, I_2, .., I_m, I_{bg}]$, where $I_{bg}$ represents the processed background images. And The training objective is to maximize the log-likelihood of predicting this sequence, where the loss is defined as:

$$\mathcal{L} = -\sum_{t=1}^{MN} \log p(q_t | q_{<t}, c) \qquad (2)$$

where $q_{<t} = (q_1, q_2, ..., q_{t-1})$. In the specific generation process, the condition $c = [c_{\mathcal{I}}, c_{\mathcal{T}}]$ is usually composed of both image and text conditions.

This strategy fully leverages the reference tokens as a dense supervisory signal to explicitly guide the model in preserving fine-grained, token-level details for each identity independently. The core of our approach is to impose a token-level constraint during the generation process. We mandate that the model's output must incorporate the complete set of tokens from all input images. This requirement acts as a direct supervisory signal, compelling the model to retain the detailed information associated with each input identity.

The efficacy of this guidance is visually substantiated in Figure 3. showing that the cross-attention maps become increasingly focused, concentrating squarely on the regions of the target objects, as progressing through the transformer layers. While attention is diffuse in the initial layers, it sharpens significantly by Layer 35, indicating that our method successfully teaches the model to isolate the spatial extent of each identity. This focused attention ensures that the generative constraints are applied precisely where needed, enabling the faithful reconstruction of identity-specific details.

### 3.1.3 INSTRUCT TOKEN INJECTION

To incorporate token-level conditioning, existing AR models typically rely on architectural modifications, such as specialized encoders or intricate feature fusion schemes. However, these approaches often treat the reference tokens as static data to be encoded, overlooking a more direct way to guide the generation process. This paradigm contrasts sharply with advancements in LLMs, where the input sequence itself is augmented with implicit task-specific guides. Inspired by the success of soft prompting methods (Wen et al., 2023) in LLMs, we introduce Instruct Token Injection. This technique introduces a set of learnable vectors that act as a dedicated container for detailed and complementary visual priors. These "instruct tokens" are prepended to the reference sequence, directly injecting

---

**Algorithm 1** Instruct Token Optimization Algorithm

**Input:** Model $\theta$, vocabulary embedding $\mathbf{E}^{|V|}$, projection function Proj, autoregressive model $\mathcal{A}$, condition encoder $\mathcal{E}$, optimization steps $T$, learning rate $\gamma$, Dataset $D$

1: Initialize token embeddings from zeros:
2: $\mathbf{P} = [\mathbf{0_i}, ...\mathbf{0_M}]$
3: **for** $1, ..., T$ **do**
4:     Retrieve current mini-batch $(X, Y) \subseteq D$.
5:     $\mathbf{X}' = \text{Proj}_{\mathbf{E}}(\mathbf{X})$
6:     Inject Instruct Embeddings:
7:     $\mathcal{X}' = \text{Concatenate}([X', P])$
8:     Calculate the gradient w.r.t. the *projected* embedding:
9:     $g = \nabla_{\mathbf{P}'} \mathcal{L}_{\text{task}}(\mathcal{A}_{\theta}(\mathcal{X}'), Y_i)$
10:    Apply the gradient on the *continuous* embedding:

11:    $\mathbf{P} = \mathbf{P} - \gamma g$
12: **end for**
13: **return** $\mathbf{P}$

---

potent, high-level guidance into the model to steer the synthesis of fine-grained details with greater precision. To endow these tokens with this capability, we optimize them directly via backpropagation. As detailed in Algorithm 1, the continuous embeddings of the instruct tokens are updated with main parameters to minimize target loss. This process effectively distills the necessary task-specific knowledge into a small set of parameters, creating powerful, reusable guides for generation.

## 4 EXPERIMENTS

### 4.1 MAIN RESULT

**Quantitative results.** We evaluate our method against other methods on SpatialSubject20K benchmark and InstructAR dataset. As shown in Table 2, our method, InstructAR (w. ITD), demonstrates superior performance, particularly on PSNR and CLIP-I metrics. However, its performance on DINO metric is slightly lower, perhaps due to a significant disparity between the poses of the reference and target objects, which leads to generation inaccuracies that may decrease the DINO score.

For multi-object preservation, we conducted evaluations on the custom-built DreamRelation dataset, comparing our approach with existing methods that support multiple reference image inputs. As

Table 2: **Quantitative Comparison on single subject reconstruction task.** Evaluations conducted on SpatialSubject200K(Wang et al., 2025a) show that our proposed method outperforms existing methods(DreamO, EditAR, InsertAnything, MIP-Adapter, MS-Diffusion). The best and the second results are demonstrated in **bold** and underlined. ↑/↓ means higher/lower is better.

| Method | Venue | Background Preservation | | Content Similarity | |
|---|---|---|---|---|---|
| | | PSNR ↑ | FID ↓ | CLIP-I ↑ | DINO ↑ |
| DreamO(Mou et al., 2025) | SIG-A 2025 | 12.30 | 102.14 | 92.11 | 84.33 |
| EditAR(Mu et al., 2025) | CVPR 2025 | 5.76 | 162.19 | 82.62 | 62.80 |
| InsertAnything(Song et al., 2025) | Arxiv 2025 | 13.88 | 103.46 | 95.36 | **91.58** |
| MIP-Adapter(Huang et al., 2024) | AAAI 2025 | 10.03 | 120.23 | 92.50 | 80.06 |
| MS-Diffusion(Wang et al., 2025b) | ICLR 2025 | 8.46 | 278.82 | 65.62 | 41.14 |
| TokenAR | Ours | **20.36** | **66.89** | **95.48** | 89.99 |

Table 3: **Quantitative Comparison on multiple subjects insertion task.** Evaluations conducted on InstructAR dataset show that our proposed method outperforms existing methods(DreamO, MIP-Adapter, MS-Diffusion) across all metrics.

| Method | Venue | Background Preservation | | Content Similarity | |
|---|---|---|---|---|---|
| | | PSNR ↑ | FID ↓ | CLIP-I ↑ | DINO ↑ |
| DreamO(Mou et al., 2025) | SIG-A 2025 | 8.62 | 145.23 | 78.65 | 63.64 |
| MIP-Adapter(Huang et al., 2024) | AAAI 2025 | 9.93 | 151.26 | 79.64 | 65.15 |
| MS-Diffusion(Wang et al., 2025b) | ICLR 2025 | 7.49 | 243.75 | 64.69 | 32.04 |
| TokenAR | Ours | **14.94** | **94.96** | **88.58** | **86.28** |

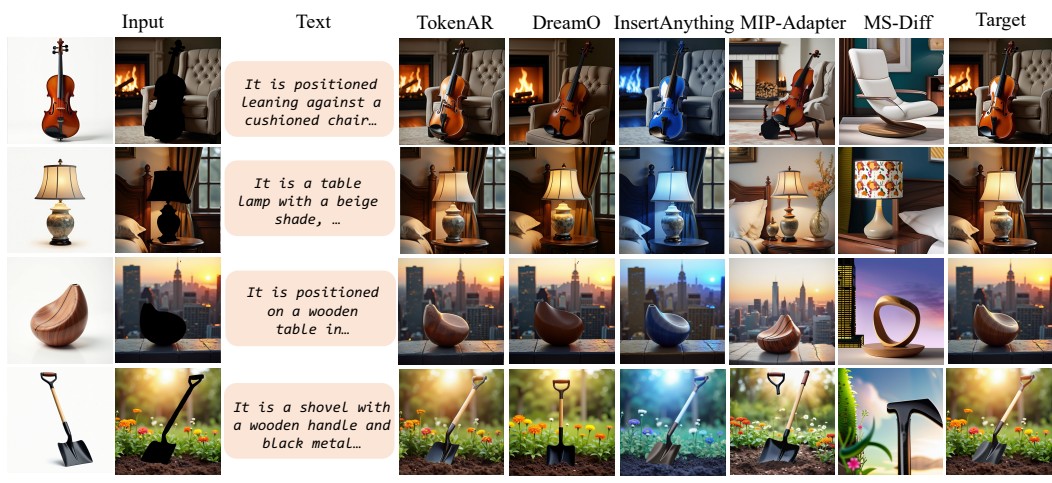

Figure 4: **Qualitative comparison on single subject insertion task.** Comparison conducted on SpatialSubject200K shows that our method **preserves identity consistency and visual coherence** in this task, with performance better than recent works.

shown in Table 3, our method outperforms existing approaches across all metrics. This success is attributed to its ability to maintain object consistency while effectively preserving the background, which significantly improves both CLIP-I and DINO scores.

**Qualitative Comparison.** Figure 4 and 5 present qualitative comparisons against mainstream baselines (e.g., DreamO, MS-Diffusion). In single-subject tasks, TokenAR excels in both identity preservation and scene integration. As visualized, our method faithfully retains the subject's intricate geometry and texture while rendering consistent lighting and shadows for seamless blending. While competing models like DreamO produce plausible outputs, they often fall short on fine details, such as rendering accurate light reflections, where TokenAR demonstrates clear superiority. For the more challenging multi-subject generation, TokenAR's key advantage is its compositional integrity. Unlike baselines that frequently suffer from identity confusion, our method successfully preserves the unique characteristics and intended relationships of each subject. Furthermore, TokenAR shows

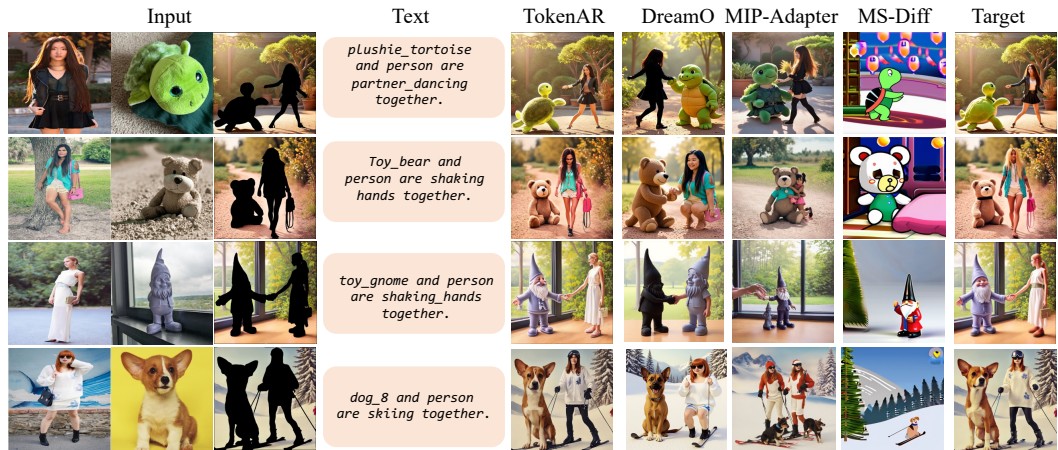

Figure 5: **Qualitative comparison on multiple subjects insertion task.** Comparison conducted on InstructAR Dataset shows that our method preserves identity consistency and visual coherence in this task, with performance better than recent works.

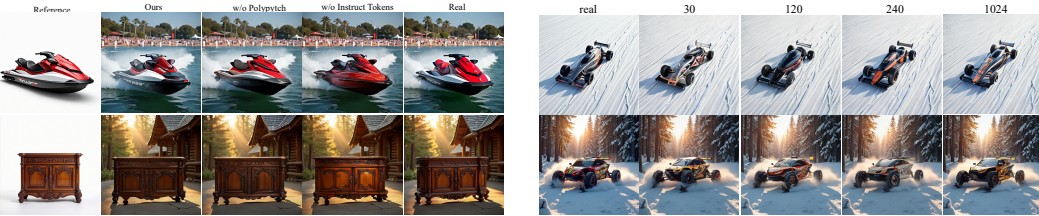

Figure 6: **Qualitative ablation study on instruct tokens mechanism and identity-token disentanglement strategy.**

Figure 7: **Qualitative ablation study on instruct token length.**

superior adherence to structural guidance from insertion masks, resulting in compositions that are significantly more coherent and visually pleasing.

## 4.2 ABLATION STUDY

Table 4: **Quantitative ablation of instruct tokens and identity-token disentanglement mechanism.**

| Method | CLIP-I ↑ | DINO ↑ |
|---|---|---|
| baseline | 90.78 | 92.30 |
| Ours(w/o Instruct) | 93.74 | 87.09 |
| Ours (w/o. ITD) | 94.77 | 87.87 |
| Ours | **95.48** | **89.99** |

**Effect of Token Index Embedding and Identity-token Disentanglement Strategy.** We conducted ablation study to verify effectiveness of our method. Figure 6 and Table 4 show that instruction tokens are crucial to maintain a high degree of visual similarity between the generated object and the original image, while the Identity-token Disentanglement Strategy company with Token Index Embedding ensures newly generated object blends seamlessly into the background. Our complete method performed best across all evaluation metrics, proving that both modules are indispensable for generating high-quality images.

**Choice of Instruct Tokens Number.** We conducted an ablation study on the length of instruct tokens to find an optimal setting. As shown in Figure 8(a), we observed a clear trade-off between CLIP Score and training loss. The CLIP Score peaked at 120 tokens, while longer sequences (e.g., 240) minimized training loss but degraded performance. An excessively long sequence (1024 tokens) proved detrimental, causing a sharp drop in CLIP Score. The cross-attention maps in Figure 8(b) provide a visual explanation for this decline: attention at 120 tokens is sharp and focused, whereas at 1024 tokens it becomes noisy and unfocused. This suggests that overly long sequences introduce disruptive noise. We therefore selected a length of 120, as it strikes the best balance between quantitative performance and focused attention.

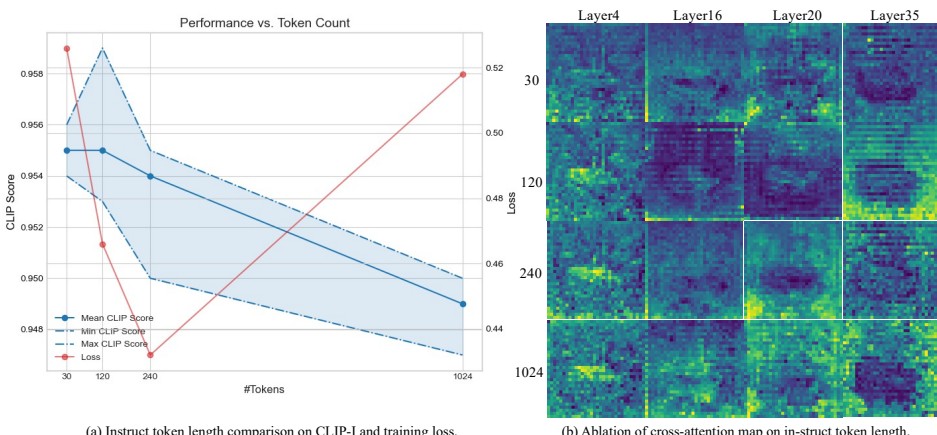

(a) Instruct token length comparison on CLIP-I and training loss.

(b) Ablation of cross-attention map on in-struct token length.

Figure 8: Ablation experiments on instruct token length.

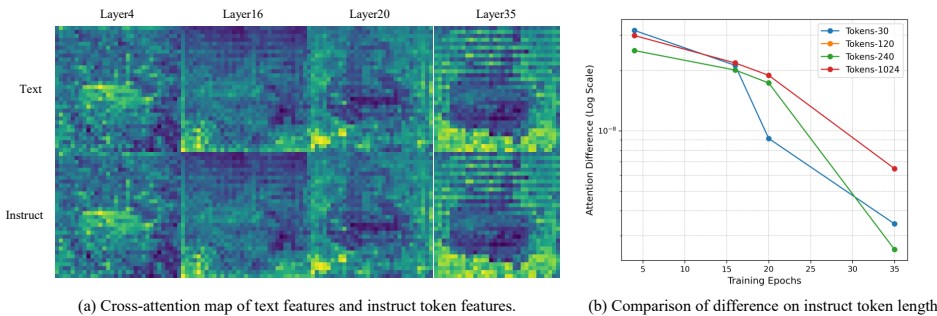

(a) Cross-attention map of text features and instruct token features.

(b) Comparison of difference on instruct token length.

Figure 9: Experiments on similarity between prompt features and instruct token features

To further investigate underlying mechanism of instruct tokens, we analyzed the similarity of cross-attention maps between instruct tokens and text prompts across Transformer layers (Figure 9). Our analysis reveals a critical dynamic: feature convergence. As information propagates through the network, the attention patterns guided by instruct tokens become progressively more focused and structured, but also increasingly similar to those of the text prompt. This is confirmed both qualitatively by the heatmaps (a) and quantitatively by the decreasing difference between their attention distributions (b). This convergence implies that in deeper layers, the feature representations of instruct tokens tend to alias with, or become redundant to, those of the text prompt. Consequently, an excessively long sequence of instruct tokens provides a diminishing and eventually detrimental signal. This insight provides a fundamental explanation for our ablation results, demonstrating that a concise sequence is optimal for preserving the distinctiveness and efficacy of the instructional guidance.

# 5    CONCLUSION

In this work, we present TokenAR, a novel autoregressive framework that resolves identity confusion in multi-subject generation. Our approach integrates three key token-level enhancements: Token Index Embedding to group tokens by origin, Instruct Token Injection to inject learnable visual priors, and an Identity-token Disentanglement strategy to foster independent feature learning. To support this task, we also introduce the InstructAR Dataset, a new large-scale benchmark. Experiments confirm that TokenAR achieves state-of-the-art performance in subject fidelity, though its sequential nature entails a trade-off in inference speed. Our work provides a robust solution and a valuable resource for advancing controllable image synthesis.

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

APPENDIX

## A PRELIMINARIES

For conditional image generation tasks using an autoregressive model, a pre-trained VQ-VAE model is typically used to encode multiple reference images into a token sequence. This model maps an input image $\mathcal{I} \in \mathbf{R}^{h \times w \times 3}$ to a vector $\mathcal{V} \in \mathbf{R}^{h' \times w' \times d}$ in the latent space, where $h$ and $w$ are the scaled height and width due to downsampling, and d is the dimension of the encoded vector. Each vector is then quantized to the most similar token in a Code Book $\mathcal{C} = \{v_k\}_{k=1}^{K}$, resulting in a token sequence $\mathbf{q} = (q_1, q_2, ..., q_N)$, where $N = h' \times w'$ and $q_i \in \{1, 2, ..., K\}$.

The discrete distribution of this token sequence, given the condition c, can be described as:

$$p(\mathbf{q}|c) = \prod_{t=1}^{N} p(q_t|q_{<t}, c) \tag{A1}$$

where $q_{<t} = (q_1, q_2, ..., q_{t-1})$. In the specific generation process, the condition $c = [c_\mathcal{I}, c_\mathcal{T}]$ is usually composed of both image and text conditions.

During the training of the model, the loss function for EditAR consists of two parts: the Cross-entropy Loss for next-token prediction and a Distill Loss. The Distill Loss is defined as:

$$\mathcal{L}_{distill} = MSE\left(\mathcal{A}\left(\mathcal{F}(\cdot)\right) \mathcal{E}_{distill}(\cdot)\right) \tag{A2}$$

where $\mathcal{A}$ is a conversion layer used only during training to align the dimensions of the Transformer features $\mathcal{F}$ with the DINO features $\mathcal{E}_{distill}$. The final training loss is a weighted sum of these two components:

$$\mathcal{L} = \mathcal{L}_{CE} + \lambda_{distill} \cdot \mathcal{L}_{distill} \tag{A3}$$

where $\mathcal{L}_{CE} = -\sum_{t=1}^{N} \log p(q_t \mid q_{<t}, c)$ is the Cross-entropy Loss.

During inference, generation is divided into two stages: prefill and decode. In the prefill stage, the model accepts the entire input, performs computations in parallel, and stores the intermediate results in a KV cache. In the decode stage, the model continues to generate each image token sequentially, following the token order.

## B SETUP

**Implementation Details.** Our method is based on EditAR(Mu et al., 2025), an image editing model based on AR model. The framework integrates a T5(Raffel et al., 2020) text encoder and the VQ-VAE encoder from LlamaGenSun et al. (2024) image encoder. For training, we set batch size of 6, with all images processed at a resolution of $512 \times 512$ pixels. We employ AdamW(Loshchilov & Hutter, 2017) optimizer with a constant learning rate of $10^{-4}$, $\beta_1 = 0.9, \beta_2 = 0.95$, applying a weigh decay of 0.05. In all experiments, we follow EditAR using $\lambda_{distill} = 0.5$ and set the maximum subject number of 4, with instruct token number of 30. All experiments are conducted on a cluster of 8 NVIDIA H20 GPUs (96GB each). We trained the model on SpatialSubject200K(Wang et al., 2025a) and InstructAR Dataset seperately, with steps of 10000 and 7000.

**Evaluation and Metrics.** To evaluate our method, we use multiple benchmark datasets. For single subject generation task, we use SpatialSubject200KWang et al. (2025a) with 200 examples. Our method uses text instruction, the subject image and masked target image to predict the target image. For multiple subject generation task, we use InstructAR Dataset, described in Sec. 2.1, which contain 200 examples. Regarding metrics, we use FID for measuring the distribution distance between generated and real images, SSIM and PSNR for evaluating structural and pixel-level similarity, LPIPS for evaluating perceptual similarity, MSE for measuring mean squared error, F1 for evaluating accuracy in object-related tasks, CLIP-I and DINO for measuring high-level semantic similarity.

Table A1: **Comparison of inference time cost and Total parameters.**

| Model | Inference Time ↓ | Total Params ↓ |
|---|---|---|
| DreamO, 2 cond | 32.30s | 12B |
| insertanything, 2 cond | 63.03s | 12B |
| MIP-Adapter, 2 cond | 8.12s | 2.6B |
| MS-Diffusion, 2 cond | **3.60s** | 2.6B |
| Ours (w/o. ITD), 2 cond | 26.43s | **0.77B** |
| Ours (w. ITD), 2 cond | 122.37s | **0.77B** |

## B.1 COMPUTATIONAL COST

In our ablation study, we evaluate the performance of our proposed method, focusing on its parameter efficiency and inference speed. As shown in Table A1, our approach has a significant advantage in the number of additionally introduced parameters. Our model introduces only 0.77B parameters, which is substantially less than all baseline models, including MIP-Adapter (2.6B) and MS-Diffusion (2.6B), as well as the much larger DreamO (12B) and InsertAnything (12B). This demonstrates that our method drastically reduces model complexity while maintaining efficient performance, allowing for a more lightweight deployment.

In terms of inference time, our method without the ITD module takes 26.43 seconds. While this is longer than some highly optimized models like MS-Diffusion, it remains within a reasonable range, especially considering its extremely low parameter count. When the ITD module is added, the inference time increases to 122.37 seconds. However, this is expected as it enables more precise control and higher-quality generation, which is acceptable for specific applications requiring high fidelity.

Overall, our method achieves an optimal balance, excelling in model parameter efficiency and maintaining a reasonable inference speed, making it particularly suitable for applications where model size and deployment costs are critical.

## C REPRODUCIBILITY STATEMENT

We have already elaborated on all the models or algorithms proposed, experimental configurations, and benchmarks used in the experiments in the main body or appendix of this paper. Furthermore, we declare that the entire code used in this work will be released after acceptance.

## D THE USE OF LARGE LANGUAGE MODELS

We use large language models solely for polishing our writing, and we have conducted a careful check, taking full responsibility for all content in this work.

## E RELATED WORK

### E.1 AUTOREGRESSIVE MODELS

AR models provide a unified framework to deal with both visual tokens and text tokens. This framework typically encodes images into token sequences through the encoder of VQ-VAE (Van Den Oord et al., 2017; Razavi et al., 2019) as tokenizer and framed generation as a next-token prediction task. VAR(Tian et al., 2024) and Parti(Yu et al., 2022) use tokens to represent image scales, while Infinity(Han et al., 2025) encodes images into bits and correct bit-wise feature. Built upon LlamaGen(Sun et al., 2024), ContextAR(Chen et al., 2025) control attention region to reconstruct image information from many kinds of conditions (e.g., canny). EditAR(Mu et al., 2025) surpasses image editing by introducing special loss function. However, these methods, designed for conditional generation, perform poorly when more conditions are provided, which demands preservation of fine-grained features in token level and during generating. Therefore, we propose learnable token index embedding and polypytch generation strategy for assisting the model to maintain high-frequency details.

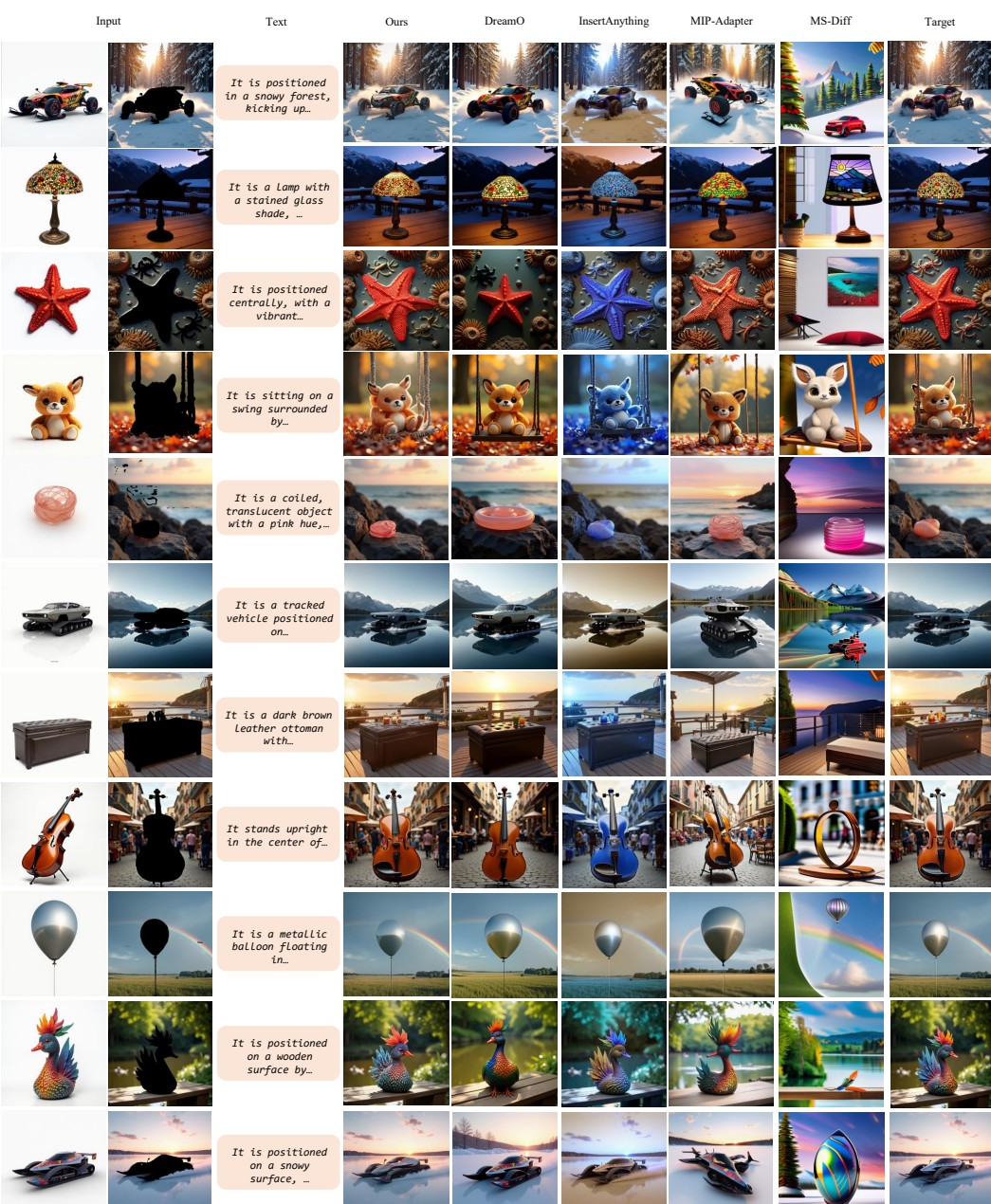

Figure A1: **Qualitative comparison on single subject insertion task.**

## E.2 MULTIPLE REFERENCE GENERATION DATASETS

Datasets for multiple reference generation task are constructed from two path, gathered from real world(Huang et al., 2024; Ruiz et al., 2022; Xiao et al., 2024) and generated by models(Wang et al., 2025a; Qingyu Shi, 2025). There are also some methods(Mou et al., 2025; Huang et al., 2024; Wang et al., 2025b; Qin et al., 2023) proposed to construct such datasets. However, these datasets lack pose transformation between input and target, and are limited to only one reference subject. For those methods provided, their mask annotations typically come from models like SAM2(Ravi et al., 2024), and BNE2(Meyer & Spruyt, 2025), which often treat the whole image as segment target regardless of reference subjects. Additionally, the filter mechanism introduced in those research only gives the core idea without any specific parameters, which makes it hard to reproduce. To address this limitation, this paper introduce the InstructAR Dataset to facilitate the training of multiple reference generation task, company with detailed setting to easily reproduce.

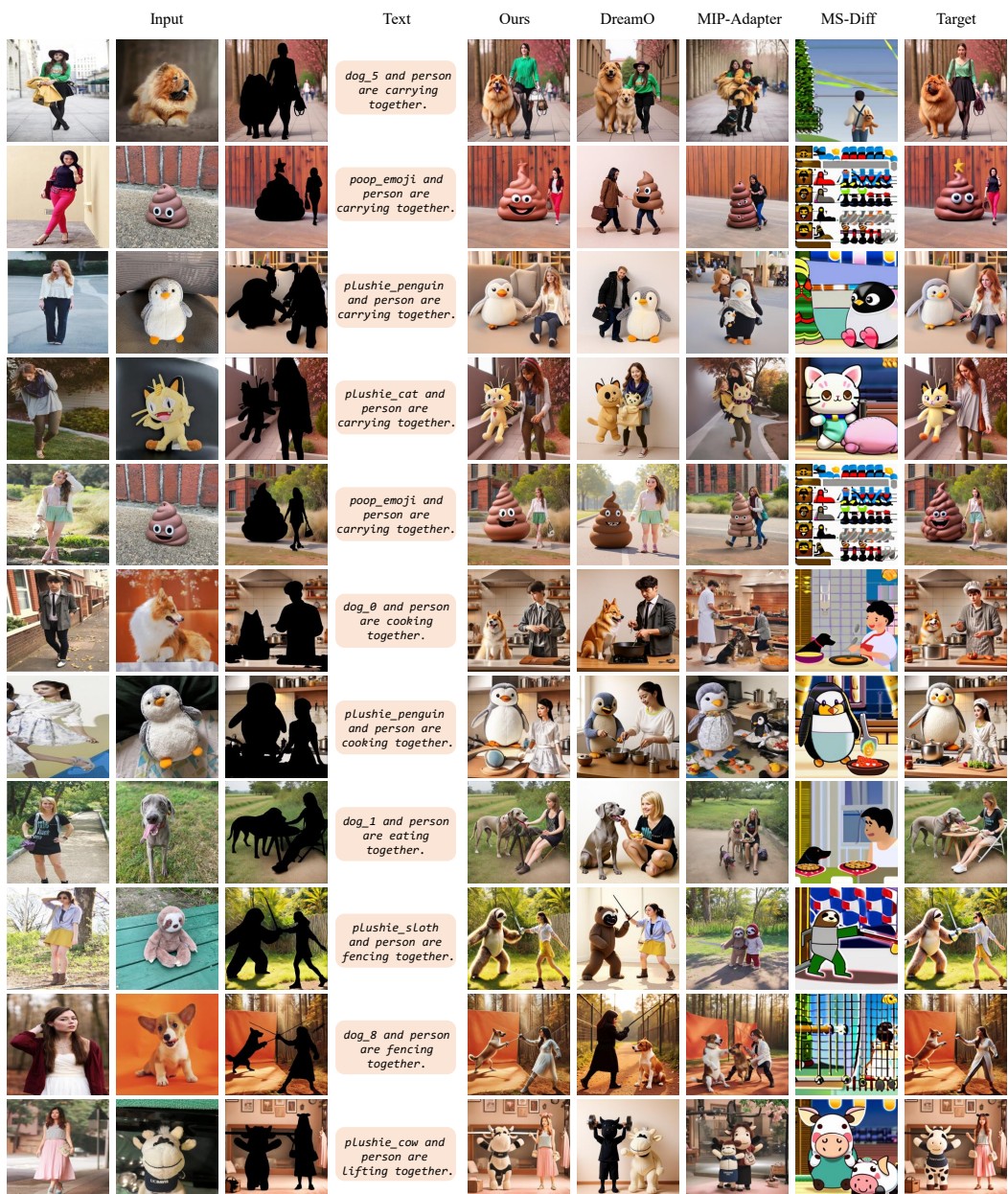

Figure A2: **Qualitative comparison on multiple subjects insertion task.**

### E.3 REFERENCE-BASED IMAGE GENERATION

Reference based image generation is focused on generation based on images and text conditions. Recent researches can be separated into two paths: 1) Diffusion based methods(Mou et al., 2025; Parmar et al., 2025; Lin et al., 2024; Mao et al., 2025; Chen et al., 2024), which integrate different types of inputs using adapter; 2) AR based methods(Chen et al., 2025; Mu et al., 2025; Shao et al., 2025), which alter the attention mechanism and construct more delicate loss function. However, most current reference-based image generation methods are limited to explore the potential capability in the model architecture itself, merely focusing on the input feature, and only support single subject generation. Our proposed InstructAR, a novel framework for multiple reference generation, introduces Instruct Token Injection method to provide more prior knowledge of the task and help distinguish multiple subject inputs during inference, while exploring the potential of trainable tokens.

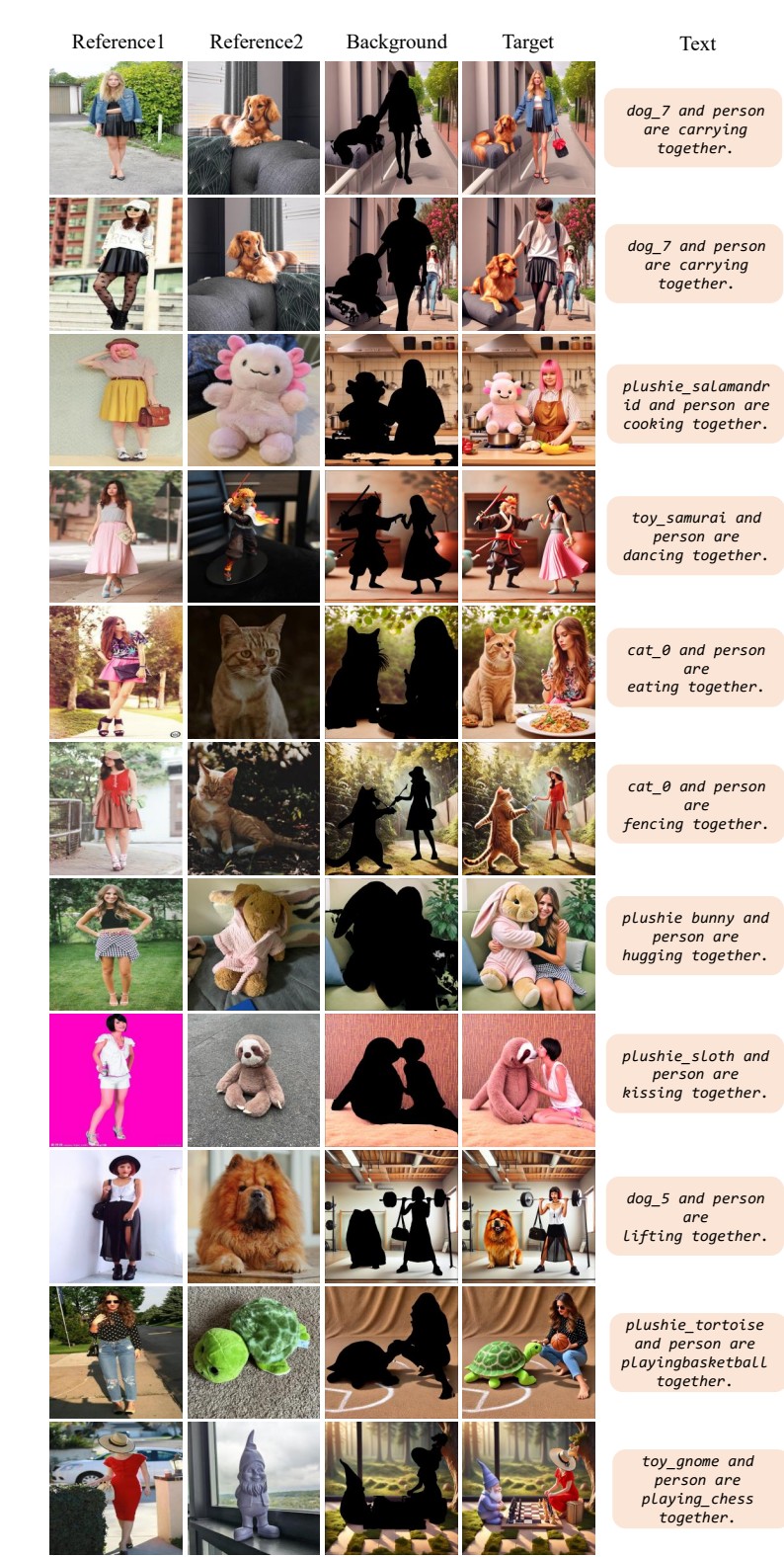

Figure A3: **Samples from InstructAR Dataset.**

