# OpenReview forum: "TokenAR: Multiple Subject Generation via Autoregressive Token-level enhancement"
_ICLR.cc/2026/Conference — ICLR 2026 Conference Withdrawn Submission_

### Official Review · Reviewer_6dKA · 2025-10-19

**Soundness:** 3
**Presentation:** 3
**Contribution:** 2
**Rating:** 4
**Confidence:** 4

**Summary:**

This paper tackles the challenging problem of multi-subject conditional image generation, where the goal is to create a coherent image featuring multiple subjects from different reference images while preserving their individual identities. The authors identify identity confusion as a key failure mode in existing autoregressive (AR) models.
To address this, they propose TokenAR, a framework built upon an AR model that introduces three novel token-level enhancements:
(1) Token Index Embedding: A learnable embedding that explicitly groups tokens originating from the same reference image, helping the model to distinguish between different subjects.
Instruct Token Injection: A set of trainable "soft prompt" tokens prepended to the input sequence to inject task-specific visual priors and guide the generation process.
(2) Identity-token Disentanglement Strategy (ITD): A training objective that compels the model to reconstruct the complete token sequence of all reference images and the background, acting as a dense supervisory signal to preserve fine-grained details for each identity.
(3) Furthermore, to facilitate research in this area, the authors introduce the InstructAR Dataset, a new large-scale, open-source dataset with 28K training pairs, specifically designed for multi-reference image generation.
The paper's contributions are thus threefold: the proposal of the TokenAR framework with its specific token-level mechanisms, the creation of the new InstructAR dataset, and extensive experiments demonstrating that TokenAR surpasses current state-of-the-art methods in both single and multi-subject generation tasks, particularly in preserving subject identity.

**Strengths:**

The paper addresses the significant and challenging problem of multi-subject conditional image generation, where maintaining subject identity is a primary difficulty. The main strengths of this work are as follows:
(1) Dataset Contribution: A significant contribution of this paper is the introduction of the InstructAR Dataset. This new, large-scale benchmark, specifically curated for multi-reference generation, is a valuable resource that can facilitate future research and standardized evaluation in this domain.
(2) Clarity: The paper is well-written and clearly structured. The motivation, methodology, and experimental setup are presented in an organized manner, making the paper easy to follow and understand.
(3) Methodological Approach: The paper presents an interesting approach by combining several token-level enhancement ideas, such as Token Index Embedding and Instruct Token Injection. The application of prompt tuning concepts (via Instruct Tokens) to this specific problem domain is a thoughtful attempt to guide the generation process.

**Weaknesses:**

Despite its interesting direction, the paper suffers from several major weaknesses that undermine the validity and significance of its contributions.
(1) Insufficient Experimental Validation on the Core Task: The paper's primary claim is solving the multi-subject generation problem. However, the critical ablation study (Table 4), which is meant to validate the contribution of each proposed component, is conducted on a single-subject task. This is a significant flaw, as it fails to provide evidence for how each component (especially Token Index Embedding and ITD) contributes to resolving identity confusion, the core challenge of the multi-subject task.
(2) Unbalanced Optimization with Severe Negative Side-effects: A deep look into the ablation results (Table 4) reveals a critical issue: while the proposed methods improve the identity similarity metric (CLIP-I), they all result in a significantly lower structural similarity score (DINO) compared to the baseline (e.g., 89.99 vs. 92.30). This strongly suggests that the optimization is a "robbing Peter to pay Paul" trade-off, where identity improvements are achieved at the cost of damaging the overall image structure and background fidelity.
(3) Prohibitive Computational Cost with Disproportionate Benefits: The core methodological contribution, the ITD strategy, demonstrates a highly unfavorable cost-benefit ratio. According to Appendix Table A1, enabling ITD increases the inference time by over 4.6 times (from 26.43s to 122.37s). However, the performance gain shown in the single-subject ablation is marginal (e.g., a ~2.4% improvement on DINO). This disproportionate trade-off severely questions the practical value and efficiency of the ITD strategy.
(4) Task Simplification and Lack of Robustness Checks: The proposed method is evaluated on a simplified task setting where the input background contains explicit "holes" (masked background), providing a very strong spatial prior for generation. The paper lacks experiments on more realistic and challenging scenarios using complete, natural backgrounds to demonstrate the method's robustness and generalization capability.
(5) Task Simplification and Unfair Comparison: The proposed method is evaluated on a simplified task setting where the input background contains explicit "holes" (masked background), providing a very strong spatial prior for generation. This raises concerns about the fairness of the comparison. To my knowledge, competing methods such as DreamO or MIP-Adapter are typically designed for more general tasks and do not necessarily rely on such strong spatial cues. If the baselines were not evaluated under the same favorable condition, the reported superiority of TokenAR could be misleading. The paper lacks experiments on more realistic scenarios using complete, natural backgrounds to demonstrate its robustness and to provide a fairer comparison against other methods.

**Questions:**

(1) Ablation Study on the Multi-Subject Task: Given that the core contribution is on multi-subject generation, can you provide a full ablation study for all proposed components (including Token Index Embedding, Instruct Tokens, and ITD) on the multi-subject InstructAR dataset? This is crucial to understand the true contribution of each component to solving the identity confusion problem.
(2) Explanation for DINO Score Degradation: Could you please explain the significant and consistent drop in the DINO score across all your proposed method variants compared to the baseline, as shown in Table 4? This suggests a negative impact on structural integrity. What steps could be taken to mitigate this issue while preserving identity?
(3) Justification for the ITD Strategy's Cost: Considering the >4x increase in inference time for what appears to be a marginal performance gain in the provided ablation, how do you justify the practical value of the ITD strategy? Have you explored any methods to achieve similar identity preservation with a more reasonable computational budget?
(4) Generalization without Strong Priors and Train-Test Inconsistency: The ITD strategy introduces a significant inconsistency between the training (reconstruction) and inference (generation) tasks. How does the model generalize so well despite this? Furthermore, how does your method perform in a more challenging setting without the strong spatial prior of a masked background, i.e., when given a complete, natural background image?

---

### Official Review · Reviewer_SxYS · 2025-10-28

**Soundness:** 3
**Presentation:** 2
**Contribution:** 3
**Rating:** 4
**Confidence:** 5

**Summary:**

To address the core issue of identity confusion in multi-subject image generation with autoregressive (AR) models, this paper proposes the TokenAR framework. It achieves multi-subject identity preservation and background fusion through three token-level enhancement components:
(1) Token Index Embedding (TIE): Clusters the token indices of the same reference image to better represent identical subjects, thereby enhancing identity consistency.
(2) Instruct Tokens: Injects structured tokens based on user instructions to guide insertion positions and semantics.
(3) Identity-token Disentanglement (ITD): Disentangles the identity features of different subjects to reduce mutual attribute contamination.
Single-subject insertion tests were conducted on SpatialSubject200K, and multi-subject insertion tests on the newly constructed InstructAR Dataset. TokenAR significantly outperforms methods such as DreamO, EditAR, InsertAnything, MIP-Adapter, and MS-Diffusion in multiple metrics including CLIP-I, DINO, and FID.

**Strengths:**

1.Targeted token-level innovations: Compared with traditional image-level or feature-level control, TokenAR’s enhancement and disentanglement at the index or position level are more fine-grained, enabling better maintenance of appearance consistency.
2.Fills data gaps: The InstructAR Dataset fills the gap in data for multi-subject interaction scenarios.
3.Sufficient experimental validation: Quantitative comparisons cover more than 5 mainstream methods, with comprehensive metric dimensions.
4.Good scalability: The token-level identity control mechanism can be ported to other AR or Diffusion generation frameworks.

**Weaknesses:**

1.Relatively artificially constructed dataset: The tasks in InstructAR are relatively structured and may not fully represent the complex scenarios of real user inputs.
2.Insufficient analysis of component interaction: Although each module has ablation results, the understanding of the interaction mechanism between Token Index Embedding, ITD, and Instruct Tokens is not in-depth enough.
3.Max subject number is set to 4, with no performance exploration for >4 subjects.
4.No standalone ablation of TIE, so its independent role in token grouping is unclear.

**Questions:**

1.What are the average inference time and GPU memory usage of TokenAR during the inference phase?
2.Can the length of Instruct Tokens be automatically determined and dynamically adjusted based on scene complexity?
3.When the number of input subjects increases to 5–6, how does the model’s performance?
4.Can you provide standalone ablation results both quantitative and qualitative for removing only TIE to demonstrate its specific contribution to identity grouping?
5.When reference images have locally similar regions, such as two same-color and same-shape cups, will TIE’s index clustering mistakenly group tokens from different subjects?
6.For ITD, how to avoid over-disentanglement, and will it cause poor fusion when a subject and background are similar in color?

---

### Official Review · Reviewer_Q48o · 2025-10-30

**Soundness:** 2
**Presentation:** 2
**Contribution:** 2
**Rating:** 2
**Confidence:** 4

**Summary:**

This paper introduces the TokenAR framework to solve identity confusion in autoregressive models for multiple reference image generation. Moreover, it collects a large-scale multi-subject dataset: InstructAR. Experiments demonstrate TokenAR outperforms state-of-the-art models in single/multi-subject generation on metrics such as PSNR and CLIP-I

**Strengths:**

1. This paper is well-written and easy to follow
2. There is sufficient work related to dataset construction and algorithm design.

**Weaknesses:**

1. The dataset lacks diversity. Although this dataset is large-scale, first, regarding the reference concept categories, there are only three types for live subjects: dog, cat, and person. Training with such limited diversity easily leads to overfitting, and the authors should rigorously analyze the model's generalization capability for other subjects. Second, concerning the relations in the dataset, "playing," which has the highest proportion, is a very ambiguous concept. Most physical contact can be interpreted as "playing," and there is a lack of relation with specific actions, such as "hugging" or "shaking."

2. The design of TokenAR is overly simplistic, and it is not clear where the innovation of this algorithm lies.

3. The design for single object insertion is simplistic. The shapes of the reference image and the segmentation map are very similar or even identical, which can be easily handled by simple ControlNet-like architectures. Have attempts been made to use different reference images? For example, replacing a violin with a guitar.

**Questions:**

Please see Weakness. I hope these issues can be resolved, and I will reconsider my grading.

---

### Official Review · Reviewer_5F3A · 2025-10-30

**Soundness:** 3
**Presentation:** 2
**Contribution:** 3
**Rating:** 6
**Confidence:** 3

**Summary:**

This work proposes a novel framework and constructs a completely new benchmark, which makes a significant contribution. However, whether ablation experiments, related work, and the new benchmark should be equipped with new evaluation metrics for a more fair assessment remains an unresolved issue.

**Strengths:**

（1）This work proposes the TokenAR framework to address the limitations of existing multiple reference generation methods, particularly the challenge of separating reference identities.

（2）To address the limitations of existing benchmarks, the InstructAR dataset has been constructed, offering further insights for establishing better and more sustainable development in this field.

**Weaknesses:**

（1）There is insufficient analysis of the proposed method. The source of harmful signals may stem from limitations within the framework itself, leading to attention degradation. The length of the tokens may not necessarily be the root cause, and further ablation studies are needed to validate the proposed hypothesis.

（2）The construction of a completely new benchmark is both necessary and contributes to the field; however, the data quality of the generated dataset has not been assessed through more specific metrics to provide more intuitive insights. Whether additional evaluation metrics should be introduced for the new benchmark to provide a more comprehensive assessment remains an unresolved issue. The sample size of the proposed dataset is significantly larger than that of other open-source benchmarks, and it remains to be verified whether the performance gains stem from the increased training data rather than the design or architectural advantages of the proposed method itself. Corresponding experiments should also be conducted on the SubjectDataset10K to validate this.

（3）The discussion of autoregressive models in related work is insufficient, as it does not address recent autoregressive models such as MAR or xAR.

**Questions:**

（1）Please provide stronger evidence that excessively long instruction token sequences yield harmful signals or discuss the potential attention degradation related to this issue[1,2].

(2) Please further explore autoregressive models such as xAR [3] or MAR [4] in the related work, and elaborate on why these autoregressive models have limitations when applied to multi-condition generation tasks, or provide relevant studies to support this argument.

Reference List：
[1] Qiu, Z., Wang, Z., Zheng, B., Huang, Z., Wen, K., Yang, S., Men, R., Yu, L., Huang, F., Huang, S., et al. Gated attention for large language models: Non-linearity, sparsity, and attention-sink-free. arXiv preprint arXiv:2505.06708, 2025.
[2] Darcet, T., Oquab, M., Mairal, J., and Bojanowski, P. Vision transformers need registers. In International Conference on Learning Representations, pp. 22466–22487, 2024.
[3] Ren, S., Yu, Q., He, J., Shen, X., Yuille, A., and Chen, L.-C. Beyond next-token: Next-x prediction for autoregressive visual generation. arXiv preprint arXiv:2502.20388, 2025.
[4] Tianhong Li, Yonglong Tian, He Li, Mingyang Deng, and Kaiming He. 2024. Autoregressive Image Generation without
 Vector Quantization. arXiv:2406.11838

---

### Official Review · Reviewer_DaAN · 2025-10-31

**Soundness:** 2
**Presentation:** 2
**Contribution:** 2
**Rating:** 2
**Confidence:** 4

**Summary:**

This paper proposes TokenAR, an autoregressive framework for multi-subject conditional image generation. To address identity confusion when multiple reference images are used, the authors introduce three token-level mechanisms: Token Index Embedding (TIE) for grouping tokens by source, Identity-Token Disentanglement (ITD) to enforce subject-specific independence, and Instruct Token Injection (ITI) to inject visual priors. A new dataset, InstructAR, is also introduced, containing over 28K annotated multi-subject samples with relational and mask information.

**Strengths:**

1. The proposed InstructAR dataset is somewhat a valuable contribution, which may benefit the overall community.
2. The proposed data construction process is simple and makes sense.

**Weaknesses:**

1. The paper is overall hard to follow.
2. The proposed dataset construction pipeline using existing methods for image generation, annotation, and filtering. It is quite similar to most prior works (e.g., OmniControl, UNO), which limits its novelty.
3. The literature review is incomplete and fails to discuss or compare with the latest approaches, e.g., UNO, Xverse.
4. The work seems to be largely empirical; the theoretical reasoning behind token-level disentanglement and identity preservation remains limited.
5. Some methodological details (e.g., the “polyptych generation strategy”) are underdefined and would benefit from clearer formalization.

**Questions:**

1. The proposed Token Index Embeddings and Identity-Token Disentanglement Strategy is not illustrated in Figure 2. Its specific implementations are not well-conveyed in the main text. How does it differ from existing token-level disentanglement methods (e.g., SAEdit, ConceptSplit)?
2. What is the precise definition of the "polyptych generation strategy"?
3. The paper lacks an analysis of computational efficiency. For instance, the ITD module appears to significantly increase inference cost, yet the efficiency trade-offs are not discussed.
4. The experimental setup is somewhat confusing. Although the paper claims to address multiple subject generation, a large portion of the experiments focuses on the subject insertion task. Moreover, the proposed method relies on precisely defined masks that accurately outline the inserted object’s shape—an unrealistic assumption for practical applications. Under such a setting, I believe that existing diffusion-based inpainting or editing methods (or exemplar-based inpainting method), combined with suitable mask strategies, could already perform well.
5. It is recommended to include a discussion on limitations and failure cases.
6. The result images presented in the paper are very small and low-resolution, making it difficult to assess image quality and subject consistency. From the limited examples provided (e.g., Figures 5 and 6), the proposed method does not clearly demonstrate an advantage in maintaining subject consistency.

---

### Note · Authors · 2025-11-12

I have read and agree with the venue's withdrawal policy on behalf of myself and my co-authors.